# Measurement of Shoulder Abduction Angle with Posture Estimation Artificial Intelligence Model

**DOI:** 10.3390/s23146445

**Published:** 2023-07-16

**Authors:** Masaya Kusunose, Atsuyuki Inui, Hanako Nishimoto, Yutaka Mifune, Tomoya Yoshikawa, Issei Shinohara, Takahiro Furukawa, Tatsuo Kato, Shuya Tanaka, Ryosuke Kuroda

**Affiliations:** Department of Orthopaedic Surgery, Kobe University Graduate School of Medicine, Kobe 650-0017, Japan; mkunose@med.kobe-u.ac.jp (M.K.); hanakoni@med.kobe-u.ac.jp (H.N.); m-ship@kf7.so-net.ne.jp (Y.M.); tomo.yoshi.0926@gmail.com (T.Y.); 203m878m@stu.kobe-u.ac.jp (I.S.); takahiro.0412.0321@gmail.com (T.F.); t.kato.ort@gmail.com (T.K.); shuyatanaka0517@gmail.com (S.T.); kurodar@med.kobe-u.ac.jp (R.K.)

**Keywords:** machine learning, MediaPipe, artificial intelligence

## Abstract

Substantial advancements in markerless motion capture accuracy exist, but discrepancies persist when measuring joint angles compared to those taken with a goniometer. This study integrates machine learning techniques with markerless motion capture, with an aim to enhance this accuracy. Two artificial intelligence-based libraries—MediaPipe and LightGBM—were employed in executing markerless motion capture and shoulder abduction angle estimation. The motion of ten healthy volunteers was captured using smartphone cameras with right shoulder abduction angles ranging from 10° to 160°. The cameras were set diagonally at 45°, 30°, 15°, 0°, −15°, or −30° relative to the participant situated at a distance of 3 m. To estimate the abduction angle, machine learning models were developed considering the angle data from the goniometer as the ground truth. The model performance was evaluated using the coefficient of determination R^2^ and mean absolute percentage error, which were 0.988 and 1.539%, respectively, for the trained model. This approach could estimate the shoulder abduction angle, even if the camera was positioned diagonally with respect to the object. Thus, the proposed models can be utilized for the real-time estimation of shoulder motion during rehabilitation or sports motion.

## 1. Introduction

The assessment of the range of motion (ROM) of the shoulder joint is crucial in the medical field for diagnosis, evaluation of disability severity, and appraisal of treatment outcomes of surgical interventions [1,2,3]. Therefore, an accurate measurement of shoulder ROM enables healthcare professionals to determine the extent of joint dysfunction and monitor the progress of treatment and rehabilitation. The universal goniometer (UG) is the most widely applied method for measuring shoulder-joint ROM in clinical settings owing to its low cost, portability, and ease of use [4,5]. However, UG measurements cannot evaluate joint angles during movement. Alternative methods, such as 3-dimensional gyroscopes [6,7], marker motion capture systems [8,9,10], inertial sensors and magnetic sensors [11,12,13,14], are limited by high costs, poor accessibility, requirement for skilled operators, and environmental constraints. Challenges in acquiring valid and repeatable data in human subjects can arise due to the relative motion and location of the skin, where markers are placed, with respect to the actual skeletal movement and location, as highlighted in previous research studies [15]. Recent advancements in computer vision and marker-less techniques have promoted the development of posture–estimation algorithms that can track human motion with high accuracy and minimal technical requirements [16,17,18,19,20,21]. As such, these algorithms can potentially revolutionize joint–angle assessments in clinical settings by overcoming the limitations of existing methods. Although still limited, there is an increasing body of literature supporting the validity of markerless motion capture systems when compared to traditional marker-based systems. For example, a study by Drazan et al. investigated lower limb angles during vertical jumps [22], while Tanaka et al. focused on lower limb joint angles during the Functional Reach Test [23]. Additionally, Wochatz et al. examined lower limb joint angles during movements including squats [24]. Nonetheless, previous studies demonstrated that camera-based posture-estimation methods entail uncertainties related to camera angles [25,26] as well as the size and ROM of body parts [27], which affect the accuracy of the joint–angle measurements. In view of the uncertainties involved in camera-based posture-estimation methods, a potential solution to this conundrum lies in the proposal to employ multiple cameras in markerless motion capture systems [28].

Within the field of clinical measurements in rehabilitation, numerous studies utilizing markerless motion capture have been conducted, many of which employ RGB-D cameras such as Microsoft Kinect [29]. RGB-D stands for “Red, Green, Blue—Depth”, referring to a type of camera that produces 3D images by combining color and distance information [30]. Similarly, reports exist regarding the use of RGB-D cameras in markerless motion capture systems for shoulder angle measurement. For instance, Gritsenko et al. [31] utilized Kinect to measure shoulder abduction in post-breast cancer surgery patients, while Beshara et al. [32] examined the reliability and validity of shoulder joint range of motion measurements by integrating wearable inertial sensors and Microsoft Kinect, with both studies reporting favorable accuracy. Moreover, with the evolution of image processing technology, depth estimation has become feasible using only RGB color imagery, enabling both tracking and machine learning tasks. This has been facilitated by a range of algorithms that allow for the recognition of human forms and the calculation of joint positions within a three-dimensional (3D) space [33]. As such, approaches employing RGB color imagery provide a more economical and practical alternative compared to methods dependent on RGB-D devices.

MediaPipe, developed by Google, is an algorithm in the domain of RGB capture. It is a universal open-source platform capable of operating on a multitude of systems. In principle, it utilizes a lightweight convolutional neural network architecture, which is specifically adjusted for real-time inference on mobile devices, for estimating 3D human posture [34]. MediaPipe BlazePose (hereafter referred to as “MediaPipe”) can evaluate the (*x*, *y*, *z*) coordinates of 33 skeletal key points for an individual from RGB images, thereby providing an attractive option for joint–angle assessments. Although MediaPipe has demonstrated superior accuracy in comparison to other posture-estimation methods, it exhibits certain limitations [35]. Existing reports suggest that MediaPipe can measure limb movements with an accuracy comparable to KinectV2 [36]; however, studies based on MediaPipe are still relatively scarce. Based on our preliminary experiments, we observed that shoulder abduction angles evaluated from coordinates detected by MediaPipe exhibited a tendency for errors. These errors increased with variations in the camera position and increasing abduction angles. These findings highlight the need for further refining of the algorithm to improve its accuracy and applicability in clinical settings. Hence, this study aimed to investigate the possibility of enhancing the detection accuracy of a shoulder-joint abduction angle by combining machine learning (ML) with the coordinate data obtained from smartphone camera images using MediaPipe for estimating the shoulder abduction angles. By addressing the limitations of the existing methods, the proposed approach aims to develop a more accurate and accessible method for assessing shoulder joint angles during motion. Therefore, this advancement is expected to improve the accuracy of diagnoses and the evaluation of treatment outcomes in patients with shoulder joint disorders, which will ultimately enhance patient care and support clinical decision-making.

## 2. Materials and Methods

### 2.1. Participants

For the assessment of right-shoulder joint angles, this study included ten healthy adult participants (five males and five females; mean age: 35 ± 5 years; mean height: 166.3 ± 8.1 cm; BMI: 22.1 ± 1.7). In particular, all participants were right-handed and volunteered in this study. The participants were instructed to perform abduction movements of the right shoulder joint in a standing position, facing forward. The researcher provided verbal instructions regarding the initial and terminal actions to be performed, and an experienced physical therapist communicated the desired actions to the volunteers. The study was approved by the Kobe University Review Board (approval number: 34261514) and informed consent was obtained from all participants.

### 2.2. Goniometric Measurements

The goniometric measurements were performed by two raters: evaluator A was an orthopedic surgeon with 8 years of clinical experience, and evaluator B was a physical therapist with 10 years of clinical experience. The participants were instructed to presume a standing position, and the measurements were performed using a Todai goniometer 200 mm (medical device notification number: 13B3X00033000015, Japan) according to the method described by Clarkson [37] for measuring the supine position (Figure 1). The participants—equipped with a strong magnetic wristband on their right hand (Figure 2)—were positioned in front of a steel wall, with their rear side in tight contact with the wall. Based on the UG measurements, the magnet was set at an angle to restrict the motion of the upper arm. Accordingly, the horizontal flexion and extension of the shoulder joint were performed at 0°, and all measurements were repeated twice at abduction angles of 10–160° in increments of 10°.

### 2.3. Data Acquisition and Image Processing by MediaPipe

After setting the shoulder-joint abduction angle, a tablet device (iPhone SE3, Apple Inc., Cupertino, CA, USA) was positioned 3 m from the participant, at a height of 150 cm above the floor. The camera was set at diagonal positions of 45°, 30°, 15°, 0°, −15°, and −30° relative to the participant standing at a distance of 3 m. In particular, the camera positioned 15° to the right of the participant was denoted as 15°, and that positioned 15° to the left was denoted as −15° (Figure 3), i.e., right and left placements were accounted as positive and negative diagonal positions, respectively. All video recordings were captured in 1080p HD at 30 fps by a designated examiner (K.M.), with each angle recorded for approximately 2 s. The video files were processed using the MediaPipe Pose Python library to obtain the joint coordinates (*x*, *y*, *z*). The *x*- and *y*-coordinates represent the horizontal and vertical coordinates from the detected hip joint center, respectively, whereas the *z*-coordinate represents the estimated distance of the object from the camera, i.e., low *z*-values indicate the proximity of the object from the camera. Among the 33 joint coordinates detected by MediaPipe [34] (Figure 4), the coordinates of the shoulder joints, hip joints, elbow joints, and nose were used for measurement. An example of an image analyzed using MediaPipe is illustrated in Figure 5, wherein the distance, angle, and area parameters were calculated using the coordinate data and vector calculations.

By following these steps, the angles, distances, and areas were evaluated using vector representations. First, a vector was created by subtracting the coordinates of the starting joint from those of the ending joint. For instance, the coordinates of the right shoulder joint were subtracted from those of the right elbow joint to construct a vector directed from the right shoulder toward the right elbow. The length of the vector is denoted by |
$a^{➔}$
| and is calculated using the Euclidean distance formula:(1)$$\left|a^{➔}\right|=\sqrt{{\left(x2-\;x1\right)}^{2}+{\left(y2-\;y1\right)}^{2}+{\left(z2-\;z1\right)}^{2}}.$$

To calculate the ratio of the vector lengths, the length of vector 
$a^{➔}$
 was divided by that of vector 
$b^{➔}$
. In principle, this ratio provides information on the relative positioning of the joints: (2)$$\mathrm{Ratio}\left(a^{➔},b^{➔}\right)=\left|a^{➔}\right|/\left|b^{➔}\right|.$$

Subsequently, the angle between vectors 
$a^{➔}$
 and 
$b^{➔}$
 was calculated using the dot-product formula and vector lengths. The arc–cosine function was used to compute the angle for a given cosine value. For calculating the 2D angles, only the *x*- and *y*-coordinates were used, excluding the *z*-coordinate.
(3)$$\mathrm{Angle}\left(a^{➔},b^{➔}\right)=\;\arccos\left(\left(a^{➔}\cdot b^{➔}\right)/\left(\left|a^{➔}\right|*\;\left|b^{➔}\right|\right)\right)$$

Here, the dot-product of 
$a^{➔}$
 · 
$b^{➔}$
 was evaluated as follows:(4)$$a^{➔}\cdot \;b^{➔}=\left(a_{x}\;*\;b_{x}\right)+\left(a_{y}{\;*\;b}_{y}\right)+\left(a_{z}{*\;b}_{z}\right).$$

Furthermore, the area between each detected coordinate was defined using the cross-product function, employing the outer product of the vectors as follows:(5)$$\mathrm{Area}\left(a^{➔},\;b^{➔}\right)=0.5\;*\;\left|\mathrm{CrossProduct}\left(a^{➔},\;b^{➔}\right)\right|,$$
where the cross-product

$a^{➔}$
 × 
$b^{➔}$
 was calculated as follows:(6)$$a^{➔}\times b^{➔}=\left(a_{x}{\;*\;b}_{y}\;\right)–\;\left(a_{y}{\;*\;b}_{x}\right).$$

### 2.4. Machine Learning (ML)

We compared the performances of the two ML algorithms—linear regression and LightGBM [38], which is a gradient boosting framework based on decision-tree learning algorithms—to estimate the shoulder abduction angle using the parameters evaluated from the estimated joint coordinates. Linear regression is a classical regression model, whereas LightGBM offers improved computational efficiency, reduced memory occupancy, and enhanced classification accuracy, while preventing overfitting. It has been used earlier to estimate hand postures from RGB images [39]. The machine-learning library Scikit-learn in Python was used for model training, and the workflow of the current experiment is illustrated in Figure 6. First, we measured the accuracy of estimating the shoulder abduction angle from the parameters derived from the images at fixed camera positions (① estimation of the shoulder abduction at the fixed camera position). Thereafter, we created a model for estimating the camera position (② estimating the camera installation position model). Following that, we incorporated the “estimate_camAngle” parameter derived from this model into the development of another model (③ estimating the shoulder abduction model at any camera installation position), which allows the detection of the shoulder abduction angle, regardless of the camera position.

In total, 66,032 images were recorded at six camera angles for 10 participants with 16 distinct shoulder abduction angles ranging from 0° to 160°. The acquired images were randomly segmented into training samples (80%) for hyperparameter tuning by generating ML models and validation samples (20%) to verify the performance of each model. After determining the optimal hyperparameters for each ML algorithm using the training samples, the coefficient of determination (R^2^), mean absolute percentage error (MAPE), and mean absolute error (MAE) were selected as performance metrics for comparing the accuracy of the employed models. The figure uses two abbreviations: Permutation feature importance and Shapley Additive exPlanations (SHAP) value. Briefly, Permutation feature importance refers to a technique for calculating the significance of different input features to the model’s predictive performance by randomly shuffling each feature and observing the effect on model accuracy. SHAP values, on the other hand, provide a measure of the contribution of each feature to the prediction for each sample, based on game theory. Detailed explanations of these terms will follow in the “Statistical analysis” section.

### 2.5. Parameters

The parameters used in the analysis, including a brief description of each parameter (Figure 7), are listed below, discussing the parameters related to the right shoulder. The parameters used for each ML model are presented in Table 1. The parameters of the faceangle and trunk (trunkAngle, trunksize) were regarded as being more indicative of the body’s direction rather than the shoulder joint angle. Consequently, they were utilized in the “Estimation of camera installation position model”. 

rtarm_distratio: The ratio of the length between the right shoulder and right elbow to that between the right shoulder and right hip joint (Figure 7: ①/②), representing the relative positional relationship of the elbow with respect to the shoulder and hip joints.rtelbowhip_distratio: The ratio of the length between the right elbow and the right hip joint to that between the right shoulder to the right hip joint (Figure 7: ③/②), reflecting the relative positional relationship of the elbow and hip joints with respect to the shoulder.rthip_distratio: The ratio of the length between the right shoulder and the right hip joint to that between the hip joints (Figure 7: ④/②), representing the relative positional relationship of the waist with respect to the shoulder.rtshoulder_distratio: The ratio of the length between the shoulder joints to that between the right shoulder and right hip joint (Figure 7: ⑤/②), clarifying the relative positional relationship of the shoulder with respect to the hip joint.rtshoulder abduction: Calculate angle ⑥ in Figure 7 from the 2D coordinates to represent the abduction angle of the right shoulder in the 2D space.rtshoulder_3Dabduction: Calculate angle ⑥ in Figure 7 from 3D coordinates to represent the abduction angle of the right shoulder in the 3D space.rtshoulderAngle: Calculate the angle ⑦ in Figure 7 from 2D coordinates to represent the angle between the right shoulder, right elbow, and right waist in the 2D space.rtshoulder_3Dangle: Calculate the angle ⑦ in Figure 7 from 3D coordinates to represent the angle between the right shoulder, right elbow, and right waist in the 3D space.rt_uppertrunkAngle: Calculate angle ⑧ in Figure 7 from 2D coordinates to represent the angle between the right shoulder, upper trunk, and left shoulder in the 2D space.lt_uppertrunkAngle: Calculate angle ⑨ in Figure 7 from 2D coordinates to represent the angle between the left shoulder, upper trunk, and right shoulder in the 2D space.rt_lowertrunkAngle: Calculate angle ⑩ in Figure 7 from 2D coordinates to represent the angle between the right waist, lower trunk, and left waist in the 2D space.lt_lowertrunkAngle: Calculate angle ⑪ in Figure 7 from 2D coordinates to represent the angle between the left waist, lower trunk, and right waist in the 2D space.rt_faceAngle: Calculate angle ⑫ in Figure 7 from 2D coordinates to represent the angle between the right side, center, and left side of the face in the 2D space.lt_faceAngle: Calculate angle ⑬ in Figure 7 from 2D coordinates to represent the angle between the left side, center, and right side of the face in the 2D space.rt_trunksize: As portrayed in Figure 7, calculate the magnitude of the cross-product of the vector from the right shoulder to the left shoulder (
$a^{➔}$
) and the vector of the length of the right trunk (
$b^{➔}$
), divided by the square of the right trunk length, representing the relative size of the right trunk area in the 2D space.


(7)
$$\mathrm{rt}\_\mathrm{trunksize}=\left|a^{➔}\times b^{➔}\right|/{\left|b^{➔}\right|}^{2}$$


lt_trunksize: As depicted in Figure 7, calculate the magnitude of the cross-product of the vector from the right shoulder to the left shoulder (
$a^{➔}$
) and the vector of the length of the left trunk (
$c^{➔}$
), divided by the square of the left trunk length, representing the relative size of the left trunk area in the 2D space.


(8)
$$\mathrm{lt}\_\mathrm{trunksize}=\left|a^{➔}\times \;c^{➔}\right|/{\left|c^{➔}\right|}^{2}$$


### 2.6. Statistical Analysis

Statistical analyses were performed using R Studio (R Studio PBC, Boston, MA, USA). The data were presented as mean values and standard deviations, and the statistical significance was indicated by *p* < 0.001. In addition, the significance value of each predictive parameter was calculated using two distinct algorithms. The significance of the permutation feature was defined as the amount by which the model score decreased upon randomly shuffling the value of a single feature. Specifically, to evaluate the significance of a certain feature, we generated a dataset with the shuffled values of the given feature, and the resulting variations in the model score were compared to the original dataset [40]. For example, the permutation feature significance of rtshoulder_distratio was calculated by shuffling its values and tallying the resulting variations in the model score. In addition, the SHAP values were defined as the contribution of each feature to the model predictions based on game theory. Furthermore, we assessed the contribution of each feature to the prediction [41]. For instance, to evaluate the impact of rt_ uppertrunkangle on the prediction, the model was trained with the remaining features, excluding rt_ uppertrunkangle, and the deviation in the model scores were evaluated. The SHAP values are insightful for understanding the significance of individual features. All ML model analyses were performed using the library Scikit-learn v1.0.2 in Python v3.8 environment.

## 3. Results

### 3.1. Estimation of Shoulder Abduction at the Fixed Camera Angle

The model was trained using the parameters listed in Table 1. In particular, shoulder_3Dabduction, shoulder abduction, and rtelbowhip_distratio exhibited strong positive correlations with the shoulder-joint abduction angle measured using the UG for each camera angle, whereas rtarm_distratio, rtshoulder_distratio, and rthip_distratio were negatively correlated. The accuracies of the ML models for each camera angle are summarized in Table 2. Compared with linear regression, LightGBM was more accurate for all camera angles. Therefore, in further experiments, we considered only the LightGBM cells.

#### 3.1.1. Estimating the Camera Installation Position Model

The model was trained using the LightGBM with the parameters listed in Table 1. As the MAPE of this model could not be evaluated, its MAE was evaluated for a performance comparison. The fixed-angle camera installation estimation model exhibited adequate accuracy, with a coefficient of determination R^2^ = 0.996 and an MAE of 0.713°.

#### 3.1.2. Estimating the Shoulder Abduction Model Irrespective of the Camera Position

As part of the explanatory data analysis (EDA), a heatmap representing the correlation between each parameter is illustrated in Figure 8. According to the heatmap, the actual angle measured by the UG was positively correlated with rtshoulder_3Dabduction, rtshoulderAbduction, and rtelbowhip_distratio, and negatively correlated with ltarm_distratio. The correlation coefficients between each parameter and the actual angle are summarized in Table 3. As all parameters were correlated with the true abduction angle, they were used to train the LightGBM model. The model performance score for the test data demonstrated a strong positive correlation between the actual angle measured by the UG and predicted values, with an R^2^ = 0.997 and an MAPE of 1.566%. To identify the significance of each parameter for predicting the shoulder abduction angle, we evaluated the feature importance. Overall, rtshoulder_3Dabduction, rtelbowhip_distratio, and rtshoulder_3Dangle were ranked as the most essential parameters in both the feature importance plot (Figure 9a) and SHAP scores (Figure 9b).

## 4. Discussion

In this study, we accurately estimated the shoulder-joint abduction angle at various camera angles by combining MediaPipe with ML models. As this paper is the first report employing such an approach, it can be deemed as novel. In the initial experiment, the camera was set at various angles to estimate the shoulder abduction angle. The camera was set at six distinct positions relative to the subject, and the shoulder-joint abduction angle was estimated using the parameters obtained from the images at each camera position combined with ML. The preliminary experiment results revealed that the error in detecting the right shoulder coordinates by the MediaPipe increased with the right shoulder abduction angle. The shouderAbduction and shoulder_3Dabduction parameters represent the angles calculated using the shoulder and hip coordinates detected by MediaPipe in 2D and 3D, respectively. Therefore, they are not equivalent to the shoulder-joint abduction angles measured using the UG. Accordingly, several parameters were adapted for model training to accurately estimate the shoulder abduction angle. In the case of diagonal positions, the center of the shoulder and waist from the RGB images could not be accurately detected, which can produce errors. However, using several parameters, we could develop an ML model with relatively high accuracy, even when the camera was placed diagonally relative to the participant. The second stage of this experiment involved estimating the camera position from the participants’ images. At this stage, the faceAngle parameter, calculated using the coordinates of the nose and the left and right shoulders, exhibited a strong correlation with the camera installation angle. In general, face detection is the most advanced technique for estimating human posture. The estimated position of the face, especially the nose, was less affected by the attire and body shape of the participant, thereby contributing to a higher accuracy than other joint adjustments. Adopting the coordinates of the facial position was highly effective for estimating the camera installation position. Third, we developed a two-stage model to estimate the shoulder abduction angle after estimating the camera installation angle. This enabled us to estimate the shoulder abduction angle without considering the camera position. The coefficient of determination, R^2^, is useful for evaluating regression analysis, and an accurate prediction is obtained when R^2^ is approximate to 1 [42]. In addition, the MAPE was used for accuracy evaluation, considering MAPE ≤ 5% = excellent match, 5% < MAPE ≤ 10% = adequate match, 10% < MAPE ≤ 15% = acceptable match, and MAPE > 15% = unacceptable match [43]. The shoulder abduction angle estimation model exhibited high accuracy, with an R^2^ = 0.997 and an MAPE of 1.539% between the angles measured by the UG and the predicted values. The precision of our proposed method gains further clarity when placed in contrast with existing literature that has evaluated shoulder abduction using markerless motion capture techniques. Beshara et al. [32] undertook an assessment of shoulder abduction using inertial sensors and Microsoft Kinect, drawing a comparison to goniometer measurements, and subsequently reported a high degree of reliability with an Intraclass Correlation Coefficient (ICC) of 0.93 and inter-rater discrepancies of no more than ±10°. Similarly, Lafayette et al. [36] appraised shoulder joint angles utilizing an RGB-D camera in conjunction with MediaPipe. Despite endorsing MediaPipe as the most accurate method in their study, they also reported an absolute deviation of 10.94° in the anterior plane and 13.87° when assessed at a 30° oblique. Conversely, in our methodology, even with an augmentation to six distinct camera placements, a high degree of accuracy was sustained with MAPE of 1.539% across ROM spanning from 0 to 160°. The positive correlations between the true_angle and rtshoulder_3DAbduction, rtshoulderAbduction, and rtelbowhip_distratio, all of which indicate the position of the right shoulder relative to the right hip and right elbow, were consistent with the positive correlation with the right shoulder joint angle. In the medical field, explainable artificial intelligence (XAI) is a collection of tools and frameworks aiming at understanding the decision-making process of ML models, while maintaining high predictive accuracy and reliability, and its significance has been emphasized in previous research [44]. Prior research incorporated SHAP and permutation feature importance analyses to ensure transparency and interpretability [40]. The permutation feature importance and SHAP scores of the current model for shoulder-joint abduction angle estimation exhibited high values for rtshoulder_3Dabduction and rtelbowhip_distraction, which represented the positions of the elbow and hip joints relative to the shoulder. These results confirmed that the angles calculated from the vectors as well as the distances between each coordinate, are crucial parameters for estimating the shoulder abduction angle using ML models. Thus, by combining posture-estimation AI MediaPipe and ML with LightGBM, the shoulder-joint abduction angle can be accurately estimated, even if the camera is positioned diagonally with respect to the participant. Therefore, a further refinement of this method enables the accurate, real-time estimation of shoulder joint movements during rehabilitation or sporting activities using RGB capture devices, which are considerably more cost-effective than RGB-D cameras. This approach thus promises to significantly enhance the accessibility and affordability of high-precision motion capture for broader applications.

### Limitations

This study posed several limitations. First, the American Academy of Orthopedic Surgeons defines the shoulder-joint abduction angle as a value between 0° and 180° [45]. In our study, we regarded a shoulder abduction angle of 160° as the upper limit because several participants could not achieve a shoulder abduction angle of 170° or 180°. Second, although the UG measurements were recorded at intervals of 10°, more precise ROM measurements may be required in clinical practice. Therefore, the extent of data should be increased by measuring the angles in smaller intervals. Third, the camera angle was adjusted from −30° to 45° in increments of 15°. One particular limitation in our study design was the absence of the −45° camera angle. The primary reason for not including this angle was that for larger-bodied participants, there was a potential that the right elbow location would not be fully captured in the image, leading to incomplete analysis. However, if this approach is applied to rehabilitation or sports motion analysis, a greater number of camera angle variations may be required, including potentially the −45° angle with necessary adjustments for larger-bodied participants. Fourth, the placement of the strong magnetic wristband on the dorsal part of the participant’s wrist likely resulted in an external rotation of the entire arm during the experiment. This could potentially affect the accuracy of our measurements, particularly when considering different physiological configurations such as placement on the ulnar styloid process. Fifth, although only the shoulder-joint abduction angle was examined, the shoulder joint can undergo various ROMs, including flexion and internal/external rotation. Therefore, the application of the current model to clinical practice may be limited for analyzing motor movements.

In future, the follow-up study will focus on further development of the proposed model with extensive data, including considering alternate wristband placement to avoid any unintentional bias in measurements, and capturing more complete ROMs for various body types and camera angles. In summary, the proposed approach, combining pose estimation AI and ML models, is advantageous for human motion analysis, despite its requirement for additional data.

## 5. Conclusions

In this study, we demonstrated the potential of employing two AI-based libraries, MediaPipe and LightGBM, for markerless motion capture and the estimation of shoulder abduction angles. Ten healthy participants were included, with shoulder abduction angles captured using smartphone cameras positioned at various diagonal angles. We utilized MediaPipe to detect the positions of key body parts such as the shoulders, elbows, hips, and nose. Additionally, we calculated the distances, angles, and areas between each joint to set the parameters accordingly. These parameters were employed as training data for the LightGBM, which yielded promising results. Moreover, considering the goniometer angle data as the ground truth, we developed ML models to estimate the abduction angle of shoulder joints. The coefficients of determination, R^2^ and MAPE, were applied for model evaluation, with the trained model yielding an R^2^ = 0.988 and an MAPE of 1.539%.

The proposed approach demonstrated the ability to estimate shoulder abduction angles even if the camera was positioned diagonally with respect to the participant. Therefore, the proposed approach has potential implications for the real-time estimation of shoulder motion during rehabilitation or sports activities. This study proposes a low-cost, high-accuracy, deep transfer learning-based image-based technique for detecting shoulder abduction angles, which exhibits a superior performance compared to conventional methods. Consequently, it enables the effective and timely estimation of shoulder abduction angles, thereby facilitating practical applications in various settings.

In conclusion, this study presents a valuable advancement in AI-based markerless motion capture for joint angle estimation. The innovative application of MediaPipe to detect body landmarks, calculate distances, angles, and areas between joints, and parameter setting for LightGBM was validated to be effective. These findings establish a solid foundation for future exploration and innovation in this field, with practical applications beyond the assessment of shoulder abduction. In future research, we envision broadening the range of joint movements assessed, such as shoulder flexion, internal and external rotation, and evaluating lower limb joint angles, by increasing the number of training angles. Additionally, we intend to apply machine learning to specific movements for more detailed motion analysis.

## Figures and Tables

**Figure 1 sensors-23-06445-f001:**
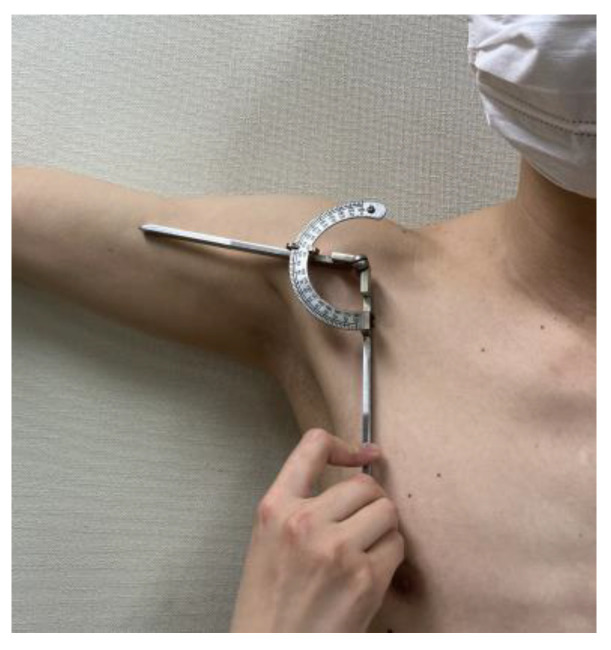
Measurement of UG.

**Figure 2 sensors-23-06445-f002:**
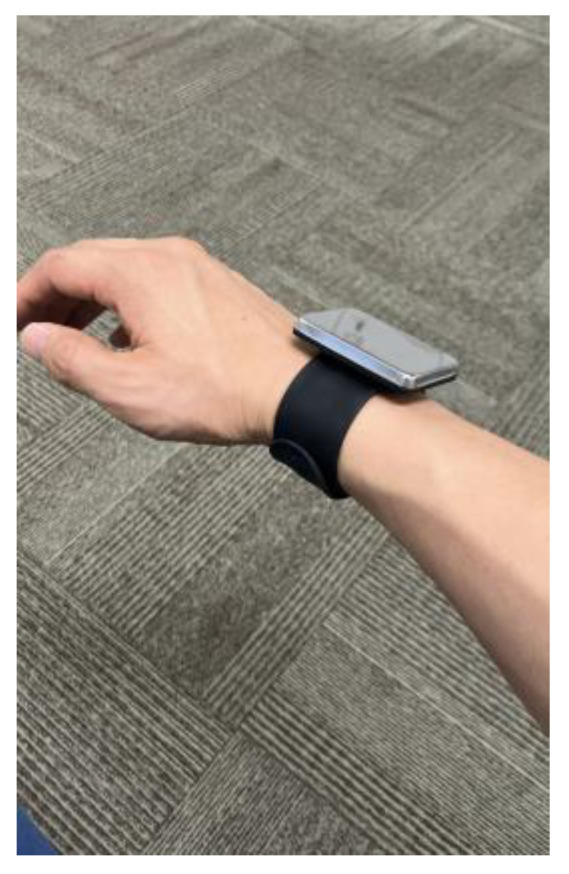
Wristband with a strong magnet.

**Figure 3 sensors-23-06445-f003:**
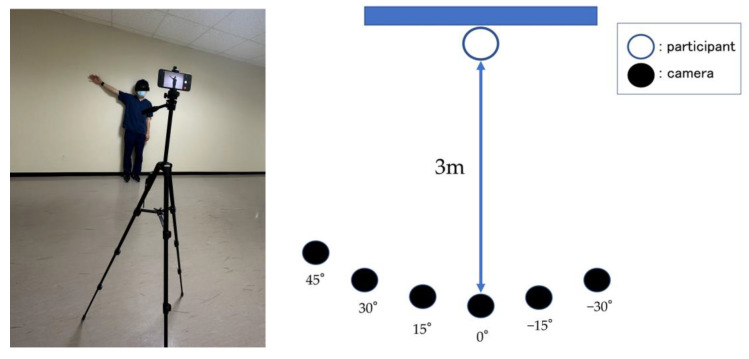
The camera position for recording.

**Figure 4 sensors-23-06445-f004:**
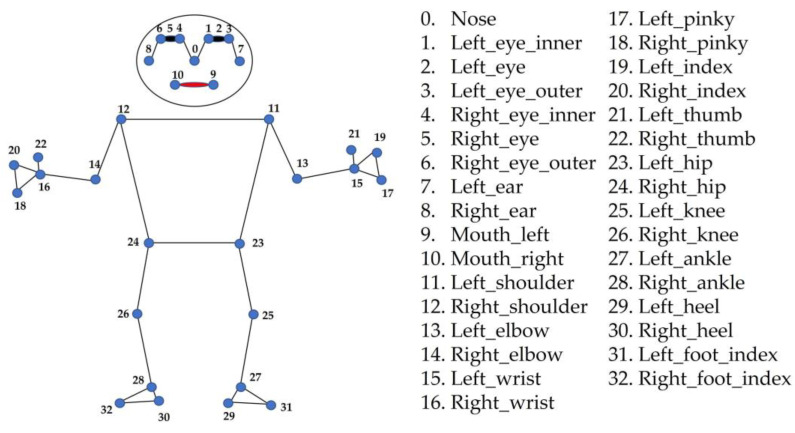
MediaPipe landmarks.

**Figure 5 sensors-23-06445-f005:**
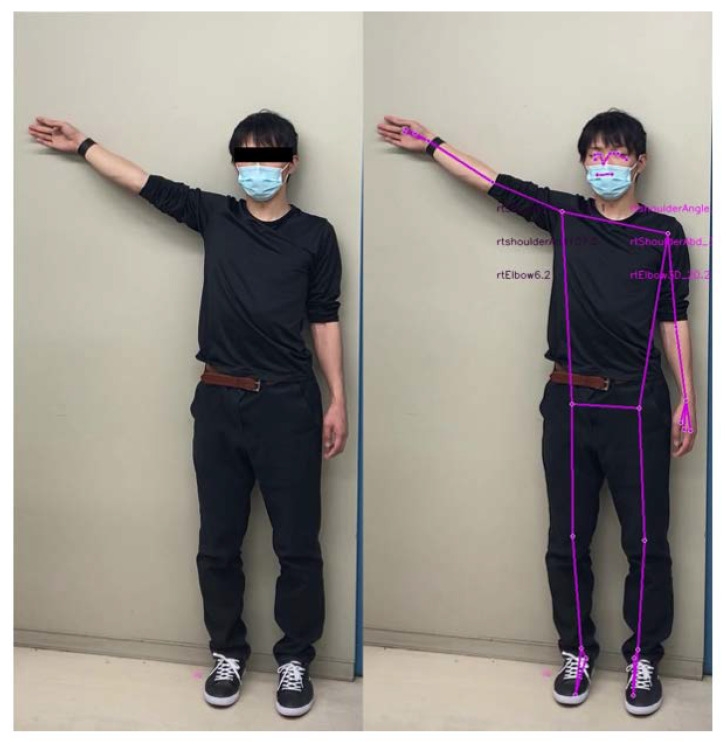
Example of joint detection by MediaPipe.

**Figure 6 sensors-23-06445-f006:**
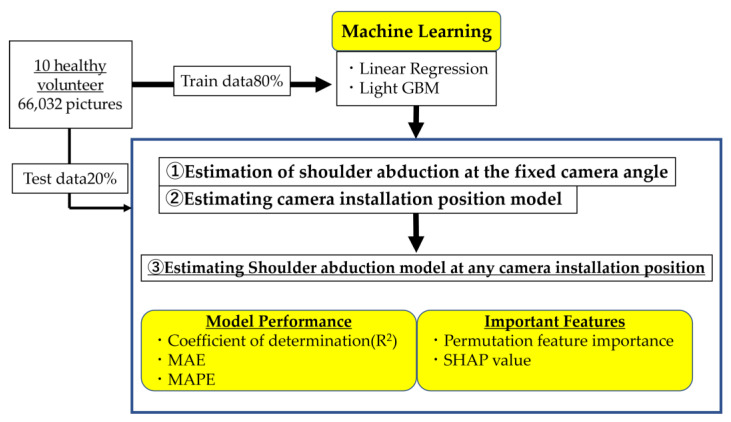
Workflow of data acquisition and machine learning.

**Figure 7 sensors-23-06445-f007:**
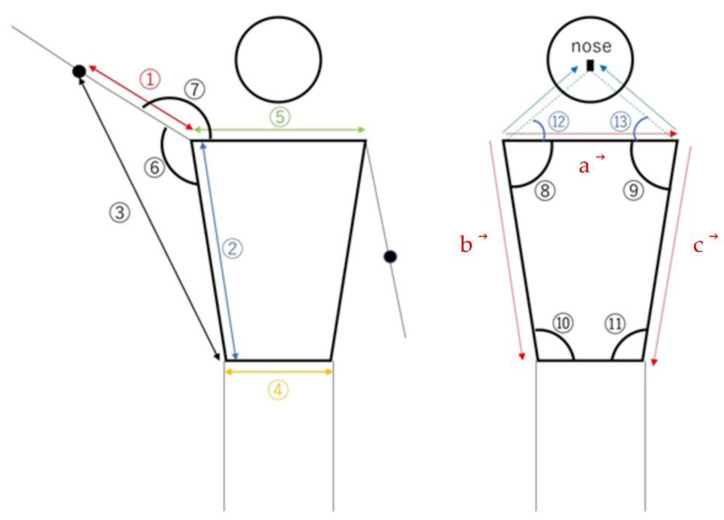
Various parameters. Numbers are explained in Table 1.

**Figure 8 sensors-23-06445-f008:**
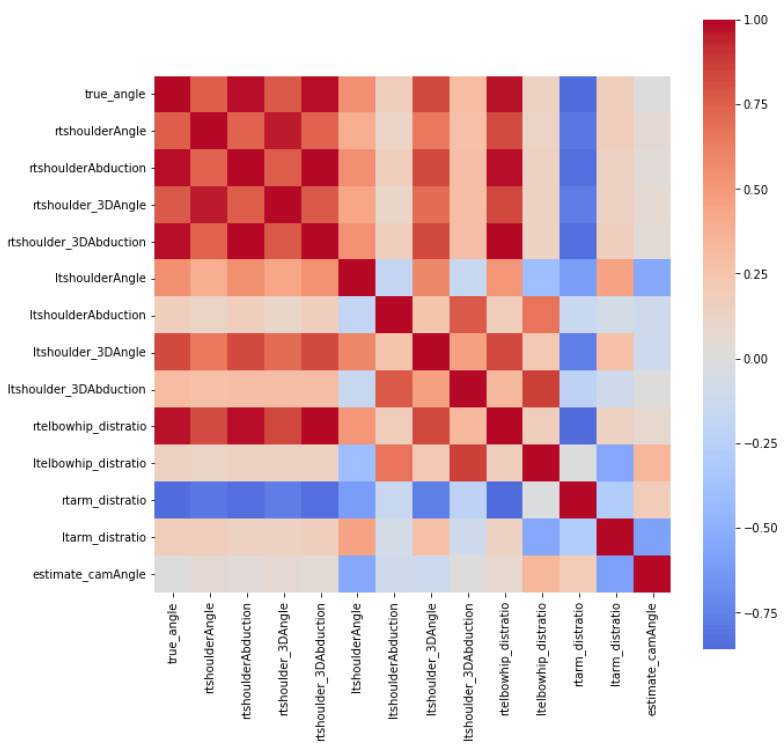
Heatmap of each parameter. Warm colors indicate a positive correlation, whereas cool colors signify a negative correlation; true_angle exhibits a positive correlation with rtshoulder_3Dabduction, rtshoulderAbduction, and rtelbowhip_distratio and a negative correlation with ltarm_distratio.

**Figure 9 sensors-23-06445-f009:**
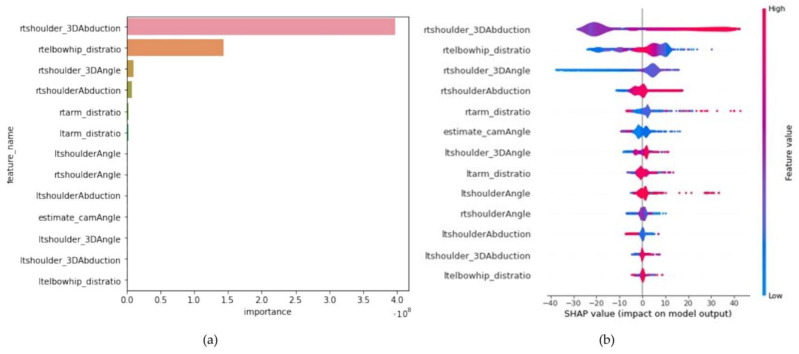
(**a**) Significance of permutation features of the LightGBM model. Top-three essential features: rtarm_distratio, rtshoulder_3Dangle, and rtshoulder_distratio. (**b**) SHAP value of the LightGBM model. Top three essential features were rtshoulder_3Dabduction, rtarm_distratio, and rtshoulder_3Dangle. Warm colors denote a positive impact on model performance, whereas cool colors indicate a negative impact.

**Table 1 sensors-23-06445-t001:** Parameters used for the training of each machine learning model.

Estimation of Shoulder Abduction at Fixed Camera Position	Estimation of Camera Installation Position Model	Estimaton of Shoulder Abduction Model at Any Camera Installation Position
arm_distratio ①/②	hip_distraio ④/②	arm_distratio ①/②
elbowhip_distratio ③/②	uppertrunkAngle ⑧, ⑨	elbowhip_distratio ③/②
hip_distraio ④/②	lowertrunkAngle ⑩, ⑪	shoulderAbduction ⑥
shoulder_distraio ⑤/②	faceAngle ⑫, ⑬	shoulder_3Dabduction ⑥
shoulderAbduction ⑥	trunksize	shoulderAngle ⑦
shoulder_3Dabduction ⑥		shoulder_3Dangle ⑦
shoulderAngle ⑦		estimate_camAngle
shoulder_3Dangle ⑦		

**Table 2 sensors-23-06445-t002:** The LightGBM model was more accurate for all camera angles.

	Camera Angle (°)					
	−30	−15	0	15	30	45
Linear regression						
correlation coefficient	0.993	0.991	0.992	0.998	0.996	0.998
R^2^	0.986	0.981	0.984	0.995	0.993	0.995
MAPE (%)	12.320	11.758	9.281	6.143	9.392	10.780
LightGBM						
correlation coefficient	1.000	1.000	1.000	1.000	0.999	0.999
R^2^	0.999	0.999	0.999	1.000	0.998	0.998
MAPE (%)	0.612	0.978	0.686	0.322	1.706	1.516

**Table 3 sensors-23-06445-t003:** Correlation coefficients of each parameter with actual angles.

Parameter	Correlation Coefficient	*p* Value
cam_angle	0.0100	<0.01
rtshoulderAngle	0.748	<0.01
rtshoulderAbduction	0.978	<0.01
rtshoulder_3Dangle	0.774	<0.01
rtshoulder_3Dabduction	0.982	<0.01
ltshoulderAngle	0.551	<0.01
ltshoulderAbduction	0.170	<0.01
ltshoulder_3Dangle	0.839	<0.01
ltshoulder_3Dabduction	0.302	<0.01
rtelbowhip_distratio	0.977	<0.01
ltelbowhip_distratio	0.133	<0.01
rtarm_distratio	−0.856	<0.01
ltarm_distratio	0.175	<0.01
rtshoulder_distratio	−0.783	<0.01
ltshoulder_distratio	0.175	<0.01
rthip_distratio	−0.860	<0.01
lthip_distratio	0.0828	<0.01

## Data Availability

The data presented in this study are available upon request from the corresponding author. The data are not publicly available because of confidentiality concerns.

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
