# Peer review of "Measurement of Shoulder Abduction Angle with Posture Estimation Artificial Intelligence Model"

_sensors, 2023, doi:10.3390/s23146445_

Round 1
Reviewer 1 Report (Previous Reviewer 2)
The manuscript seems interesting but still I think more work needs to be done;
1. What is your main contribution? Many people have proposed different AI methods for similar problens
2. Introduction ia too short. A proper literature review is needed
3. Results need to be compared with other studies.
4. I am not satisfied overall with the presence of paper. I think authors should do more work on presentation
Author Response
1. What is your main contribution? Many people have proposed different AI methods for similar problens
Response: Thank you for your feedback. Our primary contribution lies in demonstrating that the standalone use of MediaPipe for angle measurement is not optimal since the calculated angle form detected coordinates was not accurate, in this study we created the several parameters from detected coordinates and created machine learning model to estimate the shoulder abduction angle. Combination of Mediapipe and machine learning significantly enhances the precision of the angle estimation. This approach can be further leveraged to improve the accuracy of markerless motion capture systems that employ affordable RGB devices. This holds potential for wider applications, especially in instances where the cost of equipment might otherwise be prohibitive.
As we have elaborated in the updated Abstract (lines 9-11): " Substantial advancements in markerless motion capture accuracy exist, but discrepancies persist when measuring joint angles compared to those taken with a goniometer. This study integrates machine learning techniques with markerless motion capture, with an aim to enhance this accuracy.
Further, we also discuss in lines 353-357: "Therefore, further refinement of this method enables the accurate, real-time estimation of shoulder joint movements during rehabilitation or sporting activities using RGB capture devices, which are considerably more cost-effective than RGB-D cameras. This approach thus promises to significantly enhance the accessibility and affordability of high-precision motion capture for broader applications."
2. Introduction ia too short. A proper literature review is needed
Response:Thank you for your attentive review and insightful suggestions to improve our manuscript. Below, we detail the portions of the Introduction that we have revised and expanded upon.
Lines 35-38
We have supplemented the text with a citation discussing the issues of marker-based motion capture systems. Our addition reads as follows: "Challenges in acquiring valid and repeatable data in human subjects can arise due to the relative motion and location of the skin, where markers are placed, with respect to the actual skeletal movement and location, as highlighted in previous research studies [15]." (Reference 15: Morton, N.A.; Maletsky, L.P.; Pal, S.; Laz, P.J. Effect of variability in anatomical landmark location on knee kinematic description. J. Orthop. Res. 2007, 25, 1221–1230.)
Lines 42-47:
In these lines, we underscore the increasing number of reports promising results from the use of markerless motion capture systems compared to the standard marker motion capture. These reports imply the potential to bypass problems associated with marker-based systems. Here are the cited references:
[22]Drazan,J.F.; Phillips, W.T.; Seethapathi, N.; Hullfish, T.J.; Baxter, J.R. Moving outside the lab: Markerless motion capture accurately quantifies sagittal plane kinematics during the vertical jump. J. Biomech. 2021, 125, 110547.
[23]Tanaka, R.; Ishikawa, Y.; Yamasaki, T.; Diez, A. Accuracy of classifying the movement strategy in the functional reach test using a markerless motion capture system. J. Med. Eng. Technol. 2019, 43, 133–138.
[24]Wochatz, M.; Tilgner, N.; Mueller, S.; Rabe, S.; Eichler, S.; John, M.; Völler, H.; Mayer, F. Reliability and validity of the Kinect V2 for the assessment of lower extremity rehabilitation exercises. Gait. Posture 2019, 70, 330–335.
Lines 47-51:
In this section, we discuss the potential influence of camera-to-subject positioning and body size on motion analysis via markerless motion capture. We have added a citation suggesting an increase in the number of cameras to overcome the weaknesses of markerless motion capture systems and included the following sentence: "In view of the uncertainties involved in camera-based posture-estimation methods, a potential solution to this conundrum lies in the proposal to employ multiple cameras in markerless motion capture systems[28]." (Reference 28: Armitano-Lago, C.; Willoughby, D.; Kiefer, A.W. A SWOT Analysis of Portable and Low-Cost Markerless Motion Capture Systems to Assess Lower-Limb Musculoskeletal Kinematics in Sport. Front. Sports Act. Living. 2022, 3, 809898.)
We have decided to remove the following reference as we considered it inappropriate: 25. Sutherland, C.A.; Albert, W.J.; Wrigley, A.T.; Callaghan, J.P. The effect of camera viewing angle on posture assessment repeatability and cumulative spinal loading. Ergonomics 2007, 50, 877-89.
Lines 52-65:
Here, we discuss the current state of markerless motion capture, touching on RGB-D and RGB. Regarding shoulder joint angle measurement, while there are many reports using RGB-D, there are almost none using RGB. We used a cited reference on RGB-D to report the current situation. RGB-D requires a dedicated camera, but RGB capture does not require a dedicated camera, making it more economical and practical.
The following are the added texts.
In the field of clinical measurement in rehabilitation, many studies have been conducted using markerless motion capture, most of which use RGB-D cameras like Microsoft Kinect[29]. RGB-D stands for "Red, Green, Blue, Depth", a type of camera that combines color and distance information to generate 3D images[30]. Similarly, there are reports on the use of RGB-D cameras in markerless motion capture systems for shoulder angle measurement. For example, Gritsenko et al.[31] used Kinect to measure shoulder abduction in patients after breast cancer surgery, and Beshara et al.[32] integrated wearable inertial sensors and Microsoft Kinect to examine the reliability and validity of shoulder joint range of motion measurement, both reporting good accuracy. Furthermore, with the evolution of image processing technology, it has become possible to estimate depth with RGB color images only, enabling both tracking and machine learning tasks. This is facilitated by various algorithms that enable the recognition of human body shape and the calculation of joint positions in 3D space[33]. Thus, these advanced technology approaches provide more economical and practical alternatives compared to methods dependent on RGB-D devices.
The following are the references.
[31]Gritsenko V; Dailey E; Kyle N; Taylor M; Whittacre S; Swisher AK. Feasibility of using low-cost motion capture for automated screening of shoulder motion limitation after breast cancer surgery. PLoS ONE. 2015;10(6):e0128809.
[32]Beshara, P.; Chen, J.F.; Read, A.C.; Lagadec, P.; Wang, T.; Walsh, W.R. The Reliability and validity of wearable inertial sensors coupled with the microsoft kinect to measure shoulder range-of-motion. Sensors 2020, 20, 7238.
[33]Cao, Z.; Simon, T.; Wei, S.E.; Sheikh, Y. Realtime multi-person 2d pose estimation using part affinity fields. In Proceedings of the IEEE Conference on Computer Vision and Pattern Recognition, Honolulu, HI, USA, 21–26 July 2017; pp. 7291–7299.
Lines 66-75:
In this section, I added information about MediaPipe.
First, I mentioned that MediaPipe is an RGB capture explained in Lines 60-65 and that it is an open source that can be used on various systems. I also added a sentence to Lines 69-70 describing that the MediaPipe we used in this paper was MediaPipe BlazePose. There were hardly any reports on joint angle analysis by MediaPipe BlazePose, but I added the latest cited literature with the meaning of "MediaPipe can measure limb movements with an accuracy comparable to KinectV2[36], however, studies based on MediaPipe are still relatively scarce." This literature compares several RGB-D cameras and MediaPipe, but it does not include machine learning in the MediaPipe measurements and the camera position is constant. Therefore, I believe it does not overlap with our purpose.
[36]Lafayette TBG; Kunst VHL; Melo PVS; Guedes PO; Teixeira JMXN; Vasconcelos CR; Teichrieb V; da Gama AEF. Validation of Angle Estimation Based on Body Tracking Data from RGB-D and RGB Cameras for Biomechanical Assessment. Sensors (Basel). 2022 Dec 20;23(1):3
With these detailed explanations, I hope you will gain a deeper understanding of the revisions to the 'Introduction'.
3. Results need to be compared with other studies.
Response: We appreciate your feedback. As previously mentioned, our exploration revealed a limited number of studies that have reported on the use of MediaPipe. However, we have made an effort to compare our results with existing literature in lines 328-338 of the revised manuscript. We hope this amendment adequately addresses your comment. Thank you.
The following are the added texts;
lines328-338
The precision of our proposed method gains further clarity when placed in contrast with existing literature that has evaluated shoulder abduction using marker-less motion capture techniques. Beshara et al[32] undertook an assessment of shoulder abduction using inertial sensors and Mi-crosoft Kinect, drawing a comparison to goniometer measurements, and subsequently reported a high degree of reliability with an Intraclass Correlation Coefficient (ICC) of 0.93 and inter-rater discrepancies of no more than ±10°. Similarly, Lafayette et al[36] appraised shoulder joint angles utilizing an RGB-D camera in conjunction with MediaPipe. Despite endorsing MediaPipe as the most accurate method in their study, they also reported an absolute deviation of 10.94° in the an-terior plane and 13.87° when assessed at a 30° oblique. Conversely, in our methodology, even with an augmentation to six distinct camera placements, a high degree of accuracy was sustained with MAPE of 1.539% across ROM spanning from 0 to 160°.
[32]Beshara, P.; Chen, J.F.; Read, A.C.; Lagadec, P.; Wang, T.; Walsh, W.R. The Reliability and validity of wearable inertial sensors coupled with the microsoft kinect to measure shoulder range-of-motion. Sensors 2020, 20, 7238.
[36]Lafayette TBG; Kunst VHL; Melo PVS; Guedes PO; Teixeira JMXN; Vasconcelos CR; Teichrieb V; da Gama AEF. Validation of Angle Estimation Based on Body Tracking Data from RGB-D and RGB Cameras for Biomechanical Assessment. Sensors (Basel). 2022 Dec 20;23(1):3
4. I am not satisfied overall with the presence of paper. I think authors should do more work on presentation
Response:We greatly appreciate your feedback and understand your concerns regarding the presentation of the paper. We have taken your comments into consideration and made concerted efforts to improve the overall presentation, with particular attention to the areas you highlighted. We hope these revisions enhance the readability and comprehension of our work. We are grateful for your insightful feedback and patience throughout this process. Thank you once again for your guidance.
We sincerely appreciate your time and effort in reviewing our manuscript and providing valuable feedback.

Reviewer 2 Report (New Reviewer)
Dear corresponding Author,
the paper is very interesting, congratulations. The use of these methods in sports sciences or for rehabilitation purpose will be used over and over, therefore it is necessary to study in this field.
I have just a couple of comments:
1) Why you don't have the -45° camera? It should have been much more completed. Please add in the "Limitations" section
2) Regarding the wristband with strong magnet: you put it on the dorsal part of the wrist, this choice imposes an external rotation to the whole arm according to the procedures you used. Contrary, if you put the magnet on the ulnar styloid process the arm can be kept neutral during the abduction. Please add this point to the "Limitations" section.
Author Response
・Why you don't have the -45° camera? It should have been much more completed. Please add in the "Limitations" section
Response: We appreciate your insightful question.We elected not to include a -45° camera angle in our study. This decision was based on the fact that, at this particular angle, the right elbow joint was not consistently visible in the images. We foresaw that this could significantly impact the accuracy of our measurements. We are grateful for your suggestion and have now included this detail in the "Limitations" section of our manuscript.
We have added the following text to the manuscript: lines 365-371
lines365-371
A noteworthy limitation in our study design was the omission of the -45° camera angle. The primary reason for not considering this angle was that for participants with larger body sizes, there was a potential for the right elbow location not to be fully captured in the image, leading to incomplete data analysis. However, if this methodology is employed in rehabilitation or sports motion analysis, a wider variety of camera angles may be necessary, potentially including the -45° angle with appropriate adjustments for larger-bodied participants.
With this revision, we aim to address the concern raised in your comment.
Regarding the wristband with strong magnet: you put it on the dorsal part of the wrist, this choice imposes an external rotation to the whole arm according to the procedures you used. Contrary, if you put the magnet on the ulnar styloid process the arm can be kept neutral during the abduction. Please add this point to the "Limitations" section.
Response: We greatly appreciate your insightful comment. You're absolutely correct in suggesting that the placement of the wristband with a strong magnet on the ulnar styloid process could potentially maintain a neutral arm position during abduction. This observation is indeed an important consideration and has been acknowledged and added to the "Limitations" section of the paper (lines 371-374). Your guidance and patience throughout this process are highly valued. Thank you.
lines 371-374
Fourth, the placement of the strong magnetic wristband on the dorsal part of the participant's wrist likely resulted in an external rotation of the entire arm during the experiment. This could potentially affect the accuracy of our measurements, particularly when considering different physiological configurations such as placement on the ulnar styloid process.

Reviewer 3 Report (New Reviewer)
The work described in this paper is about the markerless motion capture and shoulder abduction angle estimation using two artificial intelligence-based libraries. The data of these evaluated parameters were used as training data. To estimate the abduction angle, machine learning models were developed considering the angle data from the goniometer as the ground truth. This study aimed to investigate the possibility of enhancing the detection accuracy of shoulder joint abduction angle by combining machine learning with co-ordinate data obtained from smartphone camera images for estimating shoulder abduction angles. By addressing the limitations of the existing methods, the proposed approach aims to develop a more accurate and accessible method for assessing shoulder joint angles during motion.
Some considerations regarding the content of paper:
- In the introduction, it is desirable to write in more detail about the state of research in the considered sources.
- There is too much free space on page 9. It might be better to move the information of Table 3.
- Some abbreviations are used without explanation in Figure 6.
- There are no explanations as there were selected parameters within the selected test fields in Table 2.
- There are only several references after 2020 year. What about research after this year in present time?
- It is better written also in Conclusions about future researches for this topic.
Authors should carefully examine and correct syntactic errors.
Author Response
- In the introduction, it is desirable to write in more detail about the state of research in the considered sources.
Response:Thank you for your attentive review and insightful suggestions to improve our manuscript. Below, we detail the portions of the Introduction that we have revised and expanded upon.
Lines 35-38
We have supplemented the text with a citation discussing the issues of marker-based motion capture systems. Our addition reads as follows: "Challenges in acquiring valid and repeatable data in human subjects can arise due to the relative motion and location of the skin, where markers are placed, with respect to the actual skeletal movement and location, as highlighted in previous research studies [15]." (Reference 15: Morton, N.A.; Maletsky, L.P.; Pal, S.; Laz, P.J. Effect of variability in anatomical landmark location on knee kinematic description. J. Orthop. Res. 2007, 25, 1221–1230.)
Lines 42-47:
In these lines, we underscore the increasing number of reports promising results from the use of markerless motion capture systems compared to the standard marker motion capture. These reports imply the potential to bypass problems associated with marker-based systems. Here are the cited references:
[22]Drazan,J.F.; Phillips, W.T.; Seethapathi, N.; Hullfish, T.J.; Baxter, J.R. Moving outside the lab: Markerless motion capture accurately quantifies sagittal plane kinematics during the vertical jump. J. Biomech. 2021, 125, 110547.
[23]Tanaka, R.; Ishikawa, Y.; Yamasaki, T.; Diez, A. Accuracy of classifying the movement strategy in the functional reach test using a markerless motion capture system. J. Med. Eng. Technol. 2019, 43, 133–138.
[24]Wochatz, M.; Tilgner, N.; Mueller, S.; Rabe, S.; Eichler, S.; John, M.; Völler, H.; Mayer, F. Reliability and validity of the Kinect V2 for the assessment of lower extremity rehabilitation exercises. Gait. Posture 2019, 70, 330–335.
Lines 47-51:
In this section, we discuss the potential influence of camera-to-subject positioning and body size on motion analysis via markerless motion capture. We have added a citation suggesting an increase in the number of cameras to overcome the weaknesses of markerless motion capture systems and included the following sentence: "In view of the uncertainties involved in camera-based posture-estimation methods, a potential solution to this conundrum lies in the proposal to employ multiple cameras in markerless motion capture systems[28]." (Reference 28: Armitano-Lago, C.; Willoughby, D.; Kiefer, A.W. A SWOT Analysis of Portable and Low-Cost Markerless Motion Capture Systems to Assess Lower-Limb Musculoskeletal Kinematics in Sport. Front. Sports Act. Living. 2022, 3, 809898.)
We have decided to remove the following reference as we considered it inappropriate: 25. Sutherland, C.A.; Albert, W.J.; Wrigley, A.T.; Callaghan, J.P. The effect of camera viewing angle on posture assessment repeatability and cumulative spinal loading. Ergonomics 2007, 50, 877-89.
Lines 52-65:
Here, we discuss the current state of markerless motion capture, touching on RGB-D and RGB. Regarding shoulder joint angle measurement, while there are many reports using RGB-D, there are almost none using RGB. We used a cited reference on RGB-D to report the current situation. RGB-D requires a dedicated camera, but RGB capture does not require a dedicated camera, making it more economical and practical.
The following are the added texts.
In the field of clinical measurement in rehabilitation, many studies have been conducted using markerless motion capture, most of which use RGB-D cameras like Microsoft Kinect[29]. RGB-D stands for "Red, Green, Blue, Depth", a type of camera that combines color and distance information to generate 3D images[30]. Similarly, there are reports on the use of RGB-D cameras in markerless motion capture systems for shoulder angle measurement. For example, Gritsenko et al.[31] used Kinect to measure shoulder abduction in patients after breast cancer surgery, and Beshara et al.[32] integrated wearable inertial sensors and Microsoft Kinect to examine the reliability and validity of shoulder joint range of motion measurement, both reporting good accuracy. Furthermore, with the evolution of image processing technology, it has become possible to estimate depth with RGB color images only, enabling both tracking and machine learning tasks. This is facilitated by various algorithms that enable the recognition of human body shape and the calculation of joint positions in 3D space[33]. Thus, these advanced technology approaches provide more economical and practical alternatives compared to methods dependent on RGB-D devices.
The following are the references.
[31]Gritsenko V; Dailey E; Kyle N; Taylor M; Whittacre S; Swisher AK. Feasibility of using low-cost motion capture for automated screening of shoulder motion limitation after breast cancer surgery. PLoS ONE. 2015;10(6):e0128809.
[32]Beshara, P.; Chen, J.F.; Read, A.C.; Lagadec, P.; Wang, T.; Walsh, W.R. The Reliability and validity of wearable inertial sensors coupled with the microsoft kinect to measure shoulder range-of-motion. Sensors 2020, 20, 7238.
[33]Cao, Z.; Simon, T.; Wei, S.E.; Sheikh, Y. Realtime multi-person 2d pose estimation using part affinity fields. In Proceedings of the IEEE Conference on Computer Vision and Pattern Recognition, Honolulu, HI, USA, 21–26 July 2017; pp. 7291–7299.
Lines 66-75:
In this section, I added information about MediaPipe.
First, I mentioned that MediaPipe is an RGB capture explained in Lines 60-65 and that it is an open source that can be used on various systems. I also added a sentence to Lines 69-70 describing that the MediaPipe we used in this paper was MediaPipe BlazePose. There were hardly any reports on joint angle analysis by MediaPipe BlazePose, but I added the latest cited literature with the meaning of "MediaPipe can measure limb movements with an accuracy comparable to KinectV2[36], however, studies based on MediaPipe are still relatively scarce." This literature compares several RGB-D cameras and MediaPipe, but it does not include machine learning in the MediaPipe measurements and the camera position is constant. Therefore, I believe it does not overlap with our purpose.
[36]Lafayette TBG; Kunst VHL; Melo PVS; Guedes PO; Teixeira JMXN; Vasconcelos CR; Teichrieb V; da Gama AEF. Validation of Angle Estimation Based on Body Tracking Data from RGB-D and RGB Cameras for Biomechanical Assessment. Sensors (Basel). 2022 Dec 20;23(1):3
With these detailed explanations, I hope you will gain a deeper understanding of the revisions to the 'Introduction'.
- There is too much free space on page 9. It might be better to move the information of Table 3.
Response; Thank you for pointing out the excessive space on page 9. We've revised the layout and the additional text in the introduction has effectively filled the space.
- Some abbreviations are used without explanation in Figure 6.
Response; We appreciate your careful attention to detail. We have now provided explanations for the abbreviations used in Figure 6 at lines 179-185.
The figure uses two abbreviations: Permutation feature importance and SHAP (SHapley Additive exPlanations) value. Briefly, Permutation feature importance refers to a technique for calculating the significance of different input features to the model's predictive performance by randomly shuffling each feature and observing the effect on model accuracy. SHAP values, on the other hand, provide a measure of the contribution of each feature to the prediction for each sample, based on game theory. Detailed explanations of these terms will follow in the "Statistical analysis" sec-tion.
- There are no explanations as there were selected parameters within the selected test fields in Table 2.
Response; Response: Thank you for pointing out the need for clarity regarding the selected parameters in Table 2. I wonder if you might be referring to Table 1? We considered the faceangle and trunk parameters (trunkAngle, trunksize) to be more indicative of the direction of the body rather than the shoulder joint angle. As a result, we used them in our 'Estimation of camera installation position model'. Given that we were able to obtain a reliable estimate of the camera position using these parameters in our model, we introduced a new parameter, 'estimate_camAngle', into our 'Estimation of shoulder abduction model at any camera installation position', based on the results estimated from the aforementioned model. We hope this explanation addresses your concerns and appreciate your careful attention to detail.
We have added the following text to the manuscript: lines 191-194
The parameters of the faceangle and trunk (trunkAngle, trunksize) were regarded as being more indicative of the body's direction rather than the shoulder joint angle. Consequently, they were utilized in the 'Estimation of camera installation position model'.
- There are only several references after 2020 year. What about research after this year in present time?
Response; Thank you for bringing this to our attention. We have now included additional recent references (#22, 28, 30, 32, 36) published after 2020 to provide more contemporary perspectives on the topic.
- It is better written also in Conclusions about future researches for this topic.
Response; We appreciate your suggestion. As advised, we have included further comments on potential future research avenues in lines 406-410 of the Conclusions section.
In future research, we envision broadening the range of joint movements assessed, such as shoulder flexion, internal and external rotation, and evaluating lower limb joint angles, by increasing the number of training angles. Additionally, we intend to apply machine learning to specific move-ments for more detailed motion analysis.
Comments on the Quality of English Language
Authors should carefully examine and correct syntactic errors.
Response; We acknowledge your feedback regarding the language quality. Although the manuscript had previously been reviewed by a native English speaker, we understand there were some remaining errors. Therefore, we have re-checked the grammatical errors.
We sincerely appreciate your time and effort in reviewing our manuscript and providing valuable feedback.

Round 2
Reviewer 1 Report (Previous Reviewer 2)
Paper can be accepted
This manuscript is a resubmission of an earlier submission. The following is a list of the peer review reports and author responses from that submission.
Round 1
Reviewer 1 Report
The following comments are to be addressed.
1. Line 125 - Expansion of MAPE is incorrect.
2. Equations are not referenced in text.
3. Also include more mathematical representations to lend more clarity on the concepts.
4. The quality of all figures should be improved.
5. There are many short paragraphs in the manuscripts that hamper the readability. This has to be changed.
6. Conclusion and future scope should be modified to highlight the key contributions of this work and future scope of this work should be discussed.
Overall, improve the flow of the manuscript to lend more readability.
Reviewer 2 Report
The paper is not well written. The presentation flaws are very obvious which do not give good impact to the reader.
References are not proper.
The formatting of the paper is very poor.
citations of figures and tables is not proper.
Reviewer 3 Report
Measurement of Shoulder Abduction angle with Posture Estimation Artificial Intelligence Model
Masaya Kusunose1, Atsuyuki Inui 1, Hanako Nishimoto1, ...
This manuscript proposes a solution for a practical problem using known concepts and technics. The problem is the Measurement of the Shoulder Abduction angle, while the techniques emerge from Posture Estimation using Artificial Intelligence Models.
In principle, this work can generate a good paper if the authors make some adjustments concerning the manuscript's structure:
1. Paper's scope
The manuscript is relevant in the field of applications concerning Machine Learning, especially the Measurement of the Shoulder Abduction angle.
2. Information contained
The manuscript belongs to the type "new application of known concepts."
The authors have investigated human shoulder joint abduction angles from smartphone camera images using Mediapipe. The mean absolute percentage error (MAPE) or mean absolute error (MAE) were selected as performance measures to compare the accuracy of the models.
3. The contribution
The authors claim a new practical approach for investigating human shoulder joint abduction angles from smartphone camera images using the Mediapipe system.
4. Presentation and style
Despite the good results obtained through the constructed models, the main drawback is the work's presentation:
- the lack of details concerning the analyses of the results (see below) that could help the reader reproduce some models or the newcomer that reads the paper to have an understandable example of using Machine Learning.
- the presence in the Discussion section of some text parts that do not justify their placement in this section and could be more relevant in previous sections (see below).
5. Experimental results
The results are convincing, but the Discussion section must be revised to underline only the results' interpretation.
6. Conclusions drawn
The conclusion is too poor. The authors can emphasize their contribution, how to use the developed models and the continuation of their work.
Abstract
Inside lines #17 and #125, it is written: "mean absolute error (MAPE)". It seems this must be replaced by "mean absolute percentage error" to be different from MAE.
Introduction
This section is quite a good state-of-the-art concerning the practical problem at hand, but the authors' work is not sufficiently related to the previous works. In fact, only lines #63-#65 are devoted to introducing the authors' work.
Section 2.3
The bulleted list that begins at line #134 and finishes at line #158 appears suddenly in the text. Neither part of the article nor an introducing phrase refers to or introduces this list. A simple re-lecture of the manuscript would have avoided this problem.
The authors must precede this list with an introductory text.
Section 2.4
Between lines #117 - #119, it is written:
"We compared the performance of two ML algorithms, linear regression and LightGBM, 117 which are improved gradient-boosting trees for estimating the shoulder joint abduction angle."
In my opinion, the explanations given in lines #321-#325 must be moved here, at the beginning of the manuscript, to guide the reader. IN THE DISCUSSION SECTION, USUALLY, THE AUTHORS MENTION ONLY THE NEW IDEAS RESULTING FROM THE EXPERIMENTS.
Section 2.5
In line #169, it is written:
"Permutation feature importance is defined as the amount by
which the model score decreases when one feature value is randomly shuffled".
Some details must be given to the reader eventually with an example.
The explanations in lines #326-#339 should have a better place here. I believe these lines must be moved here, at the beginning of the manuscript, to guide the reader.
In line #170, it is written:
"Shapley additive explanation (SHAP) value is defined as the contribution of each feature to the model prediction based on THE game theory".
Once again, some details must be given to the reader, accompanied eventually by an example.
For example, the explanations in lines #340-#346 should have a better place here. I believe these lines must be moved here, at the beginning of the manuscript, to guide the reader.
Congratulation on your work!
I hope my remarks will help you to obtain a good publishable article.